# STrEAMlining EFT Matching

**Timothy Cohen[1*], Xiaochuan Lu[1†] and Zhengkang Zhang[2‡]**

**1** Institute for Fundamental Science, University of Oregon, Eugene, OR 97403
**2** Walter Burke Institute for Theoretical Physics,
California Institute of Technology, Pasadena, CA 91125

★ tcohen@uoregon.edu, † xlu@uoregon.edu, ‡ zkzhang@caltech.edu

## Abstract

This paper presents STrEAM (SuperTrace Evaluation Automated for Matching), a Mathematica package that calculates all functional supertraces which arise when matching a generic UV model onto a relativistic Effective Field Theory (EFT) at one loop and to arbitrary order in the heavy mass expansion. STrEAM implements the covariant derivative expansion to automate the most tedious step of the streamlined functional matching prescription presented in Ref. [1]. The code and an example notebook are available at this link.

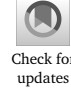

## 1 Introduction

Effective Field Theories (EFTs) provide a useful and convenient framework for describing the dynamics of low-energy degrees of freedom in a model-independent way, see *e.g.* Refs. [2–

10] for reviews. When the more fundamental UV theory is known (and calculable), one can integrate out the heavy physics and derive an EFT description valid at low energies. This so-called "matching" calculation links the Wilson coefficients in the EFT with the microscopic parameters of the UV theory.

EFT matching can be efficiently performed with functional methods [11–33]. In Ref. [1], we provided a STrEAMlined functional prescription for relativistic EFT matching up to one-loop level. Functional supertraces play an essential role; enumerating and evaluating them are the key steps in this prescription. Enumeration of the functional supertraces can be performed graphically as discussed in Ref. [1]. Evaluation can be efficiently achieved using the Covariant Derivative Expansion (CDE) technique [11–13]. This is a straightforward procedure that becomes tedious when the supertrace is complicated and/or a high operator dimension is desired. In this paper, we address this problem by introducing a `Mathematica` package, `STrEAM` (**S**uper**Tr**ace **E**valuation **A**utomated for **M**atching), which automates this procedure 🄾 [34].

Many automated tools for EFT calculations, especially in the context of the Standard Model EFT, have been developed in recent years [35–48], see Ref. [49] for a summary. Among them, `MatchingTools` [37] addresses EFT matching at tree level, while `MatchMaker` [48] (not yet released) automates Feynman diagram matching up to one-loop level. Codes are also available for partially computing one-loop EFT Lagrangians in the framework of the Universal One-Loop Effective Action [29,31,40]. To our knowledge, `STrEAM` is the first publicly available package that automates functional supertrace evaluations for general one-loop EFT matching calculations to arbitrary order in the heavy mass expansion.[1]

The rest of this paper is organized as follows. In Sec. 2, we explain the scope of `STrEAM`, *i.e.*, the specific form of functional supertraces that it evaluates; these include the two types of supertraces that appear in general one-loop functional matching calculations. In Sec. 3, we review the CDE technique and explain how to apply it to the form of supertraces targeted by `STrEAM`. We summarize the implementation of CDE in `STrEAM` and provide a simple example to demonstrate this procedure. Sec. 4 provides a user manual for `STrEAM`.

## 2 Scope of STrEAM

In this section, we discuss the motivation and specify the scope of `STrEAM`. We first review the types of functional supertraces that could arise from one-loop relativistic EFT matching calculations. We then describe the precise form of functional supertraces that `STrEAM` evaluates and explain why it covers all the possible supertraces that can appear when performing one-loop functional matching.

Consider a UV theory $\mathcal{L}_{\text{UV}}[\Phi, \phi]$ with a mass hierarchy among its fields,

$$m_\Phi \gg m_\phi \, . \tag{1}$$

We would like to integrate out the heavy fields $\Phi$ to derive an EFT for the light fields $\mathcal{L}_{\text{EFT}}[\phi]$. As elaborated in Ref. [1], one-loop matching with functional methods receives contributions from two types of supertraces:

$$\int \mathrm{d}^d x \, \mathcal{L}_{\text{EFT}}^{(\text{1-loop})}[\phi] = \frac{i}{2} \operatorname{STr} \log K \Big|_{\text{hard}} - \frac{i}{2} \sum_{n=1}^{\infty} \frac{1}{n} \operatorname{STr}\Big[\big(K^{-1} X\big)^n\Big]\Big|_{\text{hard}} \, . \tag{2}$$

We call these "log-type" and "power-type" supertraces respectively. Here "$\big|_{\text{hard}}$" means to extract the hard region contributions [51,52] in the loop integrals arising from these supertraces.

---

[1]As this project was reaching completion, we became aware of the program "SuperTracer," to be released simultaneously [50].

$K$ and $X$ are matrices acting on the space of the field multiplet

$$\varphi \equiv \begin{pmatrix} \Phi \\ \phi \end{pmatrix}. \tag{3}$$

We refer the reader to Ref. [1] for detailed definitions, as well as a discussion on how to derive $K$ and $X$ from the UV Lagrangian.

Let us first examine the log-type supertraces. In relativistic theories, the matrix $K$ has a block-diagonal form with entries

$$K_i = \begin{cases} P^2 - m_i^2 & \left(\varphi_i \text{ is spin-0}\right) \\ \slashed{P} - m_i & \left(\varphi_i \text{ is spin-}\frac{1}{2}\right) \\ -\eta^{\mu\nu}(P^2 - m_i^2) & \left(\varphi_i \text{ is spin-1}\right) \end{cases}, \tag{4}$$

where $P_\mu \equiv i D_\mu$ is the Hermitian covariant derivative. Because of the hard region requirement, only heavy field components $\varphi_i \in \{\Phi\}$ yield nonzero log-type supertraces. Up to overall constants, they are universally given by

$$i \operatorname{STr} \log K_\Phi = \begin{cases} i \operatorname{STr} \log \left(P^2 - m_\Phi^2\right) & \left(\Phi \text{ is spin-0 or spin-1}\right) \\ i \operatorname{STr} \log \left(\slashed{P} - m_\Phi\right) & \left(\Phi \text{ is spin-}\frac{1}{2}\right) \end{cases}. \tag{5}$$

The power-type supertraces, on the other hand, are not universal, since they require specifying the interaction matrix $X$. This matrix can be written as an expansion:

$$X(\phi, P_\mu) = U[\phi] + \left(P_\mu Z^\mu[\phi] + \bar{Z}^\mu[\phi] P_\mu\right) + \cdots, \tag{6}$$

where $P_\mu$ is an "open" covariant derivative. The matrix entries $U_{ij}[\phi]$, $Z_{ij}^\mu[\phi]$, $\bar{Z}_{ij}^\mu[\phi]$, *etc.* may also contain covariant derivatives, but they are closed: when viewed as operators acting on the *functional* space, where $\{|x\rangle\}$ and $\{|q\rangle\}$ each forms a basis and $\hat{x}$ and $\hat{q}$ are the basic position and momentum operators, $U_{ij}[\phi]$, $Z_{ij}^\mu[\phi]$, $\bar{Z}_{ij}^\mu[\phi]$, *etc.*, are built out of $\hat{x}$ only. In contrast, an *open* covariant derivative $P_\mu$ is built out of both $\hat{x}$ and $\hat{q}$:

$$P_\mu(\hat{x}, \hat{q}) = \hat{q}_\mu + g_a G_\mu^a(\hat{x}) T^a, \tag{7}$$

with $\hat{q}_\mu = i \partial_\mu$ in position space. In this paper, we will frequently use the notation $U_k(\hat{x})$ (or simply $U(\hat{x})$) to denote a general functional operator that is built out of $\hat{x}$ only. It can represent any of $U_{ij}[\phi]$, $Z_{ij}^\mu[\phi]$, $\bar{Z}_{ij}^\mu[\phi]$, *etc.* in Eq. (6):

$$U_k \in \left\{ U_{ij}[\phi], Z_{ij}^\mu[\phi], \bar{Z}_{ij}^\mu[\phi], \dots \right\}. \tag{8}$$

With this notation, an arbitrary term in the expansion of the matrix entry $X_{ij}$ can be expressed in the form

$$\left(P_{\mu_1} \cdots P_{\mu_n}\right) U_k \left(P_{\nu_1} \cdots P_{\nu_m}\right), \tag{9}$$

and therefore a general term in the power-type supertraces in Eq. (2),

$$-i \operatorname{STr}\left[ \frac{1}{K_{i_1}} X_{i_1 i_2} \frac{1}{K_{i_2}} X_{i_2 i_3} \cdots \frac{1}{K_{i_n}} X_{i_n i_1} \right], \tag{10}$$

is evaluated over a product sequence of segments of the form

$$\frac{1}{K_i} \left(P_{\mu_1} \cdots P_{\mu_n}\right) U_k \left(P_{\nu_1} \cdots P_{\nu_m}\right). \tag{11}$$

In what follows, we use $\Delta_i$ and $\Lambda_i$ to denote the bosonic and fermionic versions of $K_i^{-1}$, respectively:

$$\Delta_i \equiv \frac{1}{P^2 - m_i^2}, \qquad \Lambda_i \equiv \frac{1}{\slashed{P} - m_i}. \qquad (12)$$

> STrEAM automates the evaluation of functional supertraces of the form
>
> $$-i\,\mathrm{STr}\Big[ f\big(P_\mu, \{U_k\}\big) \Big]\Big|_{\mathrm{hard}}, \qquad (13)$$
>
> where $f$ is a product sequence of $P_\mu$, $U_k$, $\Delta_i$ and $\Lambda_i$, consisting of an arbitrary number of "propagator blocks":
>
> $$f = \Big[ \cdots \big(P_{\mu_1} \ldots P_{\mu_n}\big)\big(\Delta_i \text{ or } \Lambda_i\big)\big(P_{\nu_1} \ldots P_{\nu_m}\big)U_k \cdots \Big]. \qquad (14)$$

As indicated by the name, each "propagator block" has a propagator as its central object, which can be either $\Delta_i$ or $\Lambda_i$. There can be an arbitrary number of additional open covariant derivatives $P_\mu$ surrounding the propagator. A propagator block terminates with a $U$ factor, after which the next block starts. From the discussion above, it is clear that this form of $f$ covers any possible power-type supertraces.[2] We further allow the last block in $f$ to have a trivial $U$ factor, *i.e.*, $U = 1$. In this way, the log-type supertraces in Eq. (5) can also be covered upon taking a mass derivative

$$\frac{\partial}{\partial m_\Phi^2}\Big[ i\,\mathrm{STr}\log\big(P^2 - m_\Phi^2\big) \Big] = -i\,\mathrm{STr}\Big[ \frac{1}{P^2 - m_\Phi^2} \Big] = -i\,\mathrm{STr}\big[\Delta_\Phi\big]\big|_{\mathrm{hard}}, \qquad (15a)$$

$$\frac{\partial}{\partial m_\Phi}\Big[ i\,\mathrm{STr}\log\big(\slashed{P} - m_\Phi\big) \Big] = -i\,\mathrm{STr}\Big[ \frac{1}{\slashed{P} - m_\Phi} \Big] = -i\,\mathrm{STr}\big[\Lambda_\Phi\big]\big|_{\mathrm{hard}}. \qquad (15b)$$

Next, we describe the CDE method for evaluating these supertraces.

## 3   Algorithm

STrEAM implements CDE as its central algorithm to evaluate the supertraces. This technique was originally developed in Refs. [11–13], and applied to modern EFT matching calculations in Refs. [14, 15]; a variant was also developed later in Refs. [20, 22, 23]. A comprehensive review of both versions of the CDE can be found in App. B of Ref. [30], where they were termed "original CDE" and "simplified CDE" respectively. The algorithm implemented in STrEAM is the original CDE; see App. B.2.3 of Ref. [30]. In this section, we briefly review it with a focus on how it is implemented in STrEAM.

### 3.1   CDE Review

The simplified CDE and original CDE are techniques that one can use to evaluate a functional supertrace

$$-i\,\mathrm{STr}\Big[ f\big(P_\mu, \{U_k\}\big) \Big] = \int \mathrm{d}^d x \sum_i c_i\,\mathcal{O}_i(x). \qquad (16)$$

---

[2]If the last $\frac{1}{K_{i_n}} X_{i_n i_1}$ segment in Eq. (10) has $P_\mu$ factors after the $U_k$ factor (see Eq. (11)), one can cyclically permute them to the beginning of the expression, namely before $\frac{1}{K_{i_1}}$, and then use STrEAM to evaluate it.

Here $f$ is an operator in the functional space, built out of the covariant derivative $P_\mu(\hat{x}, \hat{q})$ (see Eq. (7)) and a set of functions $\{U_k(\hat{x})\}$ (as well as constants that we will suppress). In this paper, we are eventually interested in the form of $f$ evaluated by STrEAM, *i.e.*, Eq. (14). However, the method that we review in this subsection applies to a broader class of $f$, for which we only require a well-defined power expansion in its arguments $P_\mu$ and $\{U_k\}$ about $P_\mu = \{U_k\} = 0$. For example, the log-type supertraces in Eq. (5) give $f = \log\left(P^2 - m_\Phi^2\right)$ or $f = \log\left(\slashed{P} - m_\Phi\right)$; they also satisfy this condition. Nevertheless, it is useful to think of the form in Eq. (14) as a benchmark example of $f$. The general evaluation yields a set of local operators $\mathcal{O}_i$ integrated over spacetime. In the context of EFT matching, we call $\mathcal{O}_i$ "effective operators" and $c_i$ "Wilson coefficients," although we emphasize that functional supertrace evaluation can also be used to derive one-particle-irreducible effective actions more generally.

First, we address the "super" part of the functional supertrace. This part gives an overall sign $\pm$ depending on whether the diagonal entry comes from a bosonic or fermionic field $\varphi_i$:

$$-i\,\mathrm{STr}\Big[f\big(P_\mu, \{U_k\}\big)\Big] = \pm\left\{-i\,\mathrm{Tr}\Big[f\big(P_\mu, \{U_k\}\big)\Big]\right\}. \tag{17}$$

Given the form of $f$ in Eq. (14), one can determine this overall sign from the first propagator in the product sequence. If it is $\Delta_i$ ($\Lambda_i$), then the entry comes from a bosonic (fermionic) $\varphi_i$, and we should take the $+$ ($-$) sign in Eq. (17). The only exception is that a Faddeev-Popov ghost gives a propagator $\Delta_i$, but it is a fermionic field. In this case, one should take the minus sign option in Eq. (17).

Next, we deal with the more difficult "functional" part of the functional supertrace. We start out with its definition, and then make an insertion of unity in the functional space $\mathbf{1} = \int \mathrm{d}^d x \, |x\rangle \langle x|$:

$$
\begin{aligned}
-i\,\mathrm{Tr}\Big[f\big(P_\mu, \{U_k\}\big)\Big] &= -i\int \frac{\mathrm{d}^d q}{(2\pi)^d}\, \Big\langle q\Big|\,\mathrm{tr}\Big[f\big(P_\mu, \{U_k\}\big)\Big]\Big|q\Big\rangle \\
&= -i\int \mathrm{d}^d x \int \frac{\mathrm{d}^d q}{(2\pi)^d}\, \langle q|x\rangle\Big\langle x\Big|\,\mathrm{tr}\Big[f\big(P_\mu, \{U_k\}\big)\Big]\Big|q\Big\rangle \\
&= -i\int \mathrm{d}^d x \int \frac{\mathrm{d}^d q}{(2\pi)^d}\, e^{iq\cdot x}\, \mathrm{tr}\Big[f\big(P_\mu, \{U_k\}\big)\Big] e^{-iq\cdot x}.
\end{aligned}
\tag{18}
$$

In the last line, we have used

$$\langle q|x\rangle = \big(\langle x|q\rangle\big)^* = e^{iq\cdot x}. \tag{19}$$

Given that $f\big(P_\mu, \{U_k\}\big)$ has a well defined power expansion in terms of its arguments $P_\mu$ and $\{U_k\}$, we can write

$$e^{iq\cdot x} f\big(P_\mu, \{U_k\}\big) e^{-iq\cdot x} = f\big(e^{iq\cdot x} P_\mu e^{-iq\cdot x}, \{e^{iq\cdot x} U_k e^{-iq\cdot x}\}\big), \tag{20}$$

because this is true for any term in the power expansion (upon making further insertions):

$$e^{iq\cdot x}\big(ABC^{-1}\cdots\big)e^{-iq\cdot x} = \Big[\big(e^{iq\cdot x} A e^{-iq\cdot x}\big)\big(e^{iq\cdot x} B e^{-iq\cdot x}\big)\big(e^{iq\cdot x} C e^{-iq\cdot x}\big)^{-1}\cdots\Big]. \tag{21}$$

Next, we use

$$e^{iq\cdot x} P_\mu e^{-iq\cdot x} = P_\mu + q_\mu, \tag{22a}$$

$$e^{iq\cdot x} U_k e^{-iq\cdot x} = U_k, \tag{22b}$$

to find

$$-i \operatorname{Tr}\left[f\left(P_\mu, \{U_k\}\right)\right] = -i \int \mathrm{d}^d x \int \frac{\mathrm{d}^d q}{(2\pi)^d} \operatorname{tr}\left[f\left(P_\mu - q_\mu, \{U_k\}\right)\right]. \tag{23}$$

Here we have also flipped the sign of the integration variable $q_\mu$ for future convenience.

At this point, one can already make a Taylor expansion of $f\left(P_\mu - q_\mu, \{U_k\}\right)$ in terms of the covariant derivative $P_\mu$ to obtain the effective operators (truncated according to the desired operator dimension). When performing such an expansion, each effective operator will be multiplied by an expression of the loop momentum $q_\mu$, which we will call the "$q$-section" factor. Carrying out the loop integral $-i \int \frac{\mathrm{d}^d q}{(2\pi)^d}$ over the $q$-sections gives the Wilson coefficients of the effective operators. This is the "simplified CDE" method (see App. B.2.2 of Ref. [30] for more details).

In the simplified CDE, Taylor expanding $f\left(P_\mu - q_\mu, \{U_k\}\right)$ generates effective operators in which the covariant derivatives could be either open or closed. In the end, only effective operators with closed covariant derivatives will be nonzero after performing the loop integral $-i \int \frac{\mathrm{d}^d q}{(2\pi)^d}$; these effective operators are gauge singlets. Effective operators containing open covariant derivatives are not gauge singlets, as these open covariant derivatives eventually act on the unity function $\mathbb{1}$ which yields explicit gauge fields. They always drop out upon evaluating the loop integral $-i \int \frac{\mathrm{d}^d q}{(2\pi)^d}$ with dimensional regularization, because the integrands, *i.e.*, their accompanying $q$-sections, are total derivatives in $q_\mu$.

There is in fact a way to systematically avoid effective operators containing open covariant derivatives at the point of making the expansion, *i.e.*, before performing the loop integral $-i \int \frac{\mathrm{d}^d q}{(2\pi)^d}$. This is achieved by making two further insertions in Eq. (23), before and after the factor $f\left(P_\mu - q_\mu, \{U_k\}\right)$:

$$-i \operatorname{Tr}\left[f\left(P_\mu, \{U_k\}\right)\right] = -i \int \mathrm{d}^d x \int \frac{\mathrm{d}^d q}{(2\pi)^d} e^{P \cdot \frac{\partial}{\partial q}} \operatorname{tr}\left[f\left(P_\mu - q_\mu, \{U_k\}\right)\right] e^{-P \cdot \frac{\partial}{\partial q}}. \tag{24}$$

This is the "original CDE" method (see App. B.2.3 of Ref. [30] for more details). Here the second insertion is allowed because the $q_\mu$ derivatives in $e^{-P \cdot \frac{\partial}{\partial q}}$ act on the unity function $\mathbb{1}$ to yield zero, and hence only the first term in its Taylor expansion contributes:

$$e^{-P \cdot \frac{\partial}{\partial q}} \mathbb{1} = \mathbb{1}. \tag{25}$$

The first insertion in Eq. (24) is allowed because all but its first term would yield total derivatives in $q_\mu$ and hence drop out upon evaluating the loop integral $-i \int \frac{\mathrm{d}^d q}{(2\pi)^d}$ with dimensional regularization.

Using the same argument as in Eqs. (20) and (21), these insertions can be passed to the arguments of $f\left(P_\mu - q_\mu, \{U_k\}\right)$, changing them into

$$P_\mu^{\mathrm{CDE}} \equiv e^{P \cdot \frac{\partial}{\partial q}} \left(P_\mu - q_\mu\right) e^{-P \cdot \frac{\partial}{\partial q}} = -q_\mu + G_{\mu\nu}^{\mathrm{CDE}} \partial^\nu, \tag{26a}$$

$$U_k^{\mathrm{CDE}} \equiv e^{P \cdot \frac{\partial}{\partial q}} U_k e^{-P \cdot \frac{\partial}{\partial q}} = \sum_{n=0}^{\infty} \frac{1}{n!} \left(P_{\alpha_1} \cdots P_{\alpha_n} U_k\right) \partial^{\alpha_1} \cdots \partial^{\alpha_n}, \tag{26b}$$

where the quantity $G_{\mu\nu}^{\mathrm{CDE}}$ is

$$G_{\mu\nu}^{\mathrm{CDE}} \equiv -i \sum_{n=0}^{\infty} \frac{n+1}{(n+2)!} \left(P_{\alpha_1} \cdots P_{\alpha_n} F_{\mu\nu}\right) \partial^{\alpha_1} \cdots \partial^{\alpha_n}, \tag{27}$$

with $F_{\mu\nu} \equiv -i\left[P_\mu, P_\nu\right] = g_a\, G^a_{\mu\nu}\, T^a$ denoting the sum over field strengths. In Eqs. (26b) and (27), the covariant derivatives $P_{\alpha_1} \cdots P_{\alpha_n}$ in parentheses are closed; they only act on $U_k$ and $F_{\mu\nu}$, respectively. Also, starting from Eq. (26), we reserve the shorthand notation $\partial^\alpha$ for a *momentum* derivative (as opposed to the usual position derivative):

$$\partial^\alpha \equiv \frac{\partial}{\partial q_\alpha}\,. \tag{28}$$

With Eq. (26), the functional trace in Eq. (24) becomes

$$-i\,\mathrm{Tr}\Big[f\big(P_\mu, \{U_k\}\big)\Big] = -i\int \mathrm{d}^d x \int \frac{\mathrm{d}^d q}{(2\pi)^d}\,\mathrm{tr}\Big[f\big(P_\mu^{\mathrm{CDE}}, \{U_k^{\mathrm{CDE}}\}\big)\Big]. \tag{29}$$

This is the central formula for the original CDE method. From this point, we can expand $f\big(P_\mu^{\mathrm{CDE}}, \{U_k^{\mathrm{CDE}}\}\big)$ in powers of $U_k^{\mathrm{CDE}}$ and $G_{\mu\nu}^{\mathrm{CDE}}$, and then substitute in the expressions given in Eqs. (26b) and (27) to obtain the effective operators; each of them is again multiplied by a $q$-section factor. A critical feature in this expansion procedure is that all the effective operators are generated through $U_k^{\mathrm{CDE}}$ and $G_{\mu\nu}^{\mathrm{CDE}}$, in which all the covariant derivatives are already closed. This guarantees that no effective operators with open covariant derivatives will appear. Note that each closed covariant derivative (such as those in Eqs. (26b) and (27)) has operator dimension one and the field strength $F_{\mu\nu}$ has operator dimension two. So these expansions can be truncated according to the desired EFT operator dimension. Finally, we can carry out the loop integral $-i\int \frac{\mathrm{d}^d q}{(2\pi)^d}$ over the $q$-sections to obtain the Wilson coefficients.

## 3.2 CDE Within STrEAM

Now let us apply the original CDE algorithm reviewed above to the supertraces targeted by STrEAM,

$$-i\,\mathrm{STr}\Big[f\big(P_\mu, \{U_k\}\big)\Big]\Big|_{\mathrm{hard}}, \tag{30}$$

where $f\big(P_\mu, \{U_k\}\big)$ is a product sequence of $P_\mu$, $U_k$, $\Delta_i$ and $\Lambda_i$ in the form of Eq. (14). Following the central formula in the original CDE, Eq. (29), we should make the replacements

$$P_\mu \quad \rightarrow \quad P_\mu^{\mathrm{CDE}} = -q_\mu + G_{\mu\nu}^{\mathrm{CDE}}\partial^{\,\nu}, \tag{31a}$$

$$U_k \quad \rightarrow \quad U_k^{\mathrm{CDE}}, \tag{31b}$$

so that

$$\Delta_i \quad \rightarrow \quad \Delta_i^{\mathrm{CDE}} = \frac{1}{\big(P_\mu^{\mathrm{CDE}}\big)^2 - m_i^2} = \frac{1}{(-q_\mu + G_{\mu\nu}^{\mathrm{CDE}}\partial^{\,\nu})^2 - m_i^2}\,, \tag{32a}$$

$$\Lambda_i \quad \rightarrow \quad \Lambda_i^{\mathrm{CDE}} = \frac{1}{\slashed{P}^{\mathrm{CDE}} - m_i} = \frac{1}{-\slashed{q} + \gamma^\mu G_{\mu\nu}^{\mathrm{CDE}}\partial^{\,\nu} - m_i}\,. \tag{32b}$$

Instead of using Eq. (32b), in STrEAM we adopt an alternative strategy to address fermionic propagators; we convert them into bosonic propagators:

$$\Lambda_i = \frac{1}{\slashed{P} - m_i} = \frac{1}{\slashed{P}^2 - m_i^2}(\slashed{P} + m_i) = \frac{1}{P^2 - m_i^2 - \Sigma}(\slashed{P} + m_i)$$

$$= \big(\Delta_i + \Delta_i\,\Sigma\,\Delta_i + \Delta_i\,\Sigma\,\Delta_i\,\Sigma\,\Delta_i + \cdots\big)(\slashed{P} + m_i)\,. \tag{33}$$

Here the "dipole" factor

$$\Sigma \equiv -\frac{1}{2}\sigma^{\mu\nu}F_{\mu\nu}, \qquad \text{with} \qquad \sigma^{\mu\nu} \equiv \frac{i}{2}[\gamma^\mu, \gamma^\nu], \tag{34}$$

can be viewed as a specific case of $U_k$, and it has operator dimension two. The expansion in Eq. (33) can be truncated according to the desired operator dimension.

After converting all the fermionic propagators $\Lambda_i$ to the bosonic ones $\Delta_i$ through Eq. (33), we replace $P_\mu$, $U_k$ (including $\Sigma$), and $\Delta_i$ with their CDE counterparts $P_\mu^{\text{CDE}}$, $U_k^{\text{CDE}}$, and $\Delta_i^{\text{CDE}}$ as given in Eqs. (31) and (32a). We then make use of Eqs. (26b) and (27) to expand the expression into a sum of effective operators, each accompanied by a $q$-section. For the supertraces in STrEAM, the $U_k$ factors originate from the entries $U_{ij}[\phi]$, $Z_{ij}^\mu[\phi]$, $\bar{Z}_{ij}^\mu[\phi]$, *etc.* in the matrix $X$ (see Eq. (6)), or from the dipole factor $\Sigma$ in Eq. (33); they have at least one power of the light fields and hence a minimum operator dimension one (but could be higher). Comparing the desired EFT operator dimension with the sum of those from $U_k$, we can determine where to truncate the expansion.

Finally, we perform the loop integral over the $q$-sections to obtain the Wilson coefficients. Carrying out all the $q$-derivatives and making the symmetrization of the type

$$q^{\mu_1}q^{\mu_2} \to \frac{1}{d}q^2\eta^{\mu_1\mu_2}, \tag{35a}$$

$$q^{\mu_1}q^{\mu_2}q^{\mu_3}q^{\mu_4} \to \frac{1}{d(d+2)}q^4\left(\eta^{\mu_1\mu_2}\eta^{\mu_3\mu_4} + \eta^{\mu_1\mu_3}\eta^{\mu_2\mu_4} + \eta^{\mu_1\mu_4}\eta^{\mu_2\mu_3}\right), \tag{35b}$$

one can bring the $q$-sections to (a sum of) the following form

$$\frac{\left(q^2\right)^r}{\left(q^2 - m_1^2\right)^{n_1}\left(q^2 - m_2^2\right)^{n_2}\cdots\left(q^2 - m_k^2\right)^{n_k}}. \tag{36}$$

To obtain the hard region contributions, we need to expand this integrand into a series assuming the loop momentum $q \sim m_{\text{heavy}} \gg m_{\text{light}}$. Practically, this means identifying the light masses $m_{\text{light}}$ in Eq. (36), and expanding the light propagators as

$$\frac{1}{q^2 - m_{\text{light}}^2} = \frac{1}{q^2} + \frac{m_{\text{light}}^2}{q^4} + \frac{m_{\text{light}}^4}{q^6} + \cdots. \tag{37}$$

We truncate this expansion based on the desired total power of $m_{\text{light}}$ in the Wilson coefficients. After making this hard region expansion and truncation, the integrand can be again organized into (a sum of) the form of Eq. (36), but now with only heavy propagators remaining. So in the end, all the hard region loop integrals are reduced to the general form

$$\frac{1}{16\pi^2}\text{LoopI}_{(n_1,\cdots,n_k)}^{(r)}\left(m_1^2,\cdots,m_k^2\right) \equiv -i\int\frac{\mathrm{d}^d q}{(2\pi)^d}\frac{\left(q^2\right)^r}{\left(q^2 - m_1^2\right)^{n_1}\cdots\left(q^2 - m_k^2\right)^{n_k}}, \tag{38}$$

which we then evaluate with dimensional regularization and the $\overline{\text{MS}}$ scheme. When there is no heavy mass propagator left in Eq. (38), the integral is scaleless and yields zero. When there are one or two distinct heavy masses, STrEAM provides the explicit integration results. In cases where there are three or more non-degenerate heavy masses, STrEAM leaves the loop integral in the abstract form that appears on the left hand side of Eq. (38).

## 3.3 Implementation Summary

In `STrEAM`, a functional supertrace

$$-i\,\mathrm{STr}\Big[f\big(P_\mu,\{U_k\}\big)\Big]\Big|_{\mathrm{hard}},\tag{39}$$

with $f\big(P_\mu,\{U_k\}\big)$ a product sequence of $P_\mu$, $U_k$, $\Delta_i$ and $\Lambda_i$ in the form of Eq. (14),

$$f = \Big[\ \cdots\ \big(P_{\mu_1}\dots P_{\mu_n}\big)\big(\Delta_i\ \text{or}\ \Lambda_i\big)\big(P_{\nu_1}\dots P_{\nu_m}\big)U_k\ \cdots\ \Big],\tag{40}$$

is evaluated with the following steps:

- **Address the "super" in STr.** Assign an overall sign $+$ $(-)$ as in Eq. (17) if the first propagator is $\Delta_i$ $(\Lambda_i)$. Keep in mind that a ghost field propagator is an exception to this rule for which one needs an extra overall minus sign.

- **Address fermionic propagators.** Apply the relation in Eq. (33) to convert all the fermionic propagators $\Lambda_i$ into bosonic propagators $\Delta_i$. Truncate the expansion according to the desired operator dimension in the EFT.

- **Perform original CDE.** Apply the central formula of original CDE, Eq. (29), where $P_\mu$, $U_k$ (including $\Sigma$), and $\Delta_i$ are replaced with their CDE counterparts $P_\mu^{\mathrm{CDE}}$, $U_k^{\mathrm{CDE}}$, and $\Delta_i^{\mathrm{CDE}}$ given in Eqs. (31) and (32a). Then make use of Eqs. (26b) and (27) to expand the expression into a sum of effective operators, each multiplied by a $q$-section factor (a function of $q$). Truncate the expansion according to the desired operator dimension in the EFT.

- **Perform loop integrals.** For each effective operator, simplify its accompanying $q$-section into (a sum of) the form of Eq. (36) by carrying out the $q$-derivatives and making the symmetrization of the type in Eq. (35). Expand and truncate the light propagators as in Eq. (37). Compute the resulting integrals in the form of Eq. (38) to obtain the Wilson coefficients.

## 3.4 A Simple Example

In this subsection, we work out a simple example by hand as a pedagogical demonstration of the evaluation procedure in Sec. 3.3. We evaluate the supertrace

$$T_1 \equiv -i\,\mathrm{STr}\bigg[\frac{1}{P^2-m_1^2}U_1^{[2]}\bigg]\bigg|_{\mathrm{hard}} = -i\,\mathrm{STr}\Big[\Delta_1 U_1^{[2]}\Big]\Big|_{\mathrm{hard}},\tag{41}$$

assuming that $m_1$ is a heavy mass (otherwise the hard region contribution vanishes). We evaluate this up to operator dimension six. Following the notation in Ref. [1], we have put a superscript "[2]" on $U_1$ to indicate that its minimum operator dimension is two.

**Address the "super" in STr**

In this example, the "super" part gives a positive sign because the first propagator $\Delta_1$ is a bosonic propagator (except when it comes from a ghost):

$$T_1 = -i\,\mathrm{Tr}\Big[\Delta_1 U_1^{[2]}\Big]\Big|_{\mathrm{hard}}.\tag{42}$$

**Address fermionic propagators**

There is no fermionic propagator $\Lambda_i$ to address in this example.



**Perform original CDE**

Following the central formula of the original CDE, Eq. (29), we make the replacements in Eqs. (31) and (32a) to obtain

$$
T_1 = \left[ -i \int d^d x \int \frac{d^d q}{(2\pi)^d} \, \mathrm{tr}\big(\Delta_1^{\mathrm{CDE}} U_1^{\mathrm{CDE}}\big) \right]\bigg|_{\mathrm{hard}}.
\tag{43}
$$

Now we need to use Eqs. (26b) and (27) to expand this expression into a sum of effective operators truncated at dimension six.

First we expand the factor $\Delta_1^{\mathrm{CDE}}$. Because we desire a result up to operator dimension six and $U_1^{\mathrm{CDE}}$ already has operator dimension two, we only need to expand $\Delta_1^{\mathrm{CDE}}$ up to operator dimension four, which is at most two powers of $G_{\mu\nu}^{\mathrm{CDE}}$:

$$
\begin{aligned}
\Delta_1^{\mathrm{CDE}} &= \frac{1}{\left(-q_\mu + G_{\mu\nu}^{\mathrm{CDE}}\partial^\nu\right)^2 - m_1^2} \\
&= \frac{1}{q^2 - m_1^2 - \left[\left(q^\mu G_{\mu\nu}^{\mathrm{CDE}} + G_{\mu\nu}^{\mathrm{CDE}} q^\mu\right)\partial^\nu - \eta^{\mu\nu} G_{\mu\alpha}^{\mathrm{CDE}} G_{\nu\beta}^{\mathrm{CDE}} \partial^\alpha\partial^\beta\right]} \\
&= \frac{1}{q^2 - m_1^2} + \frac{1}{q^2 - m_1^2}\left(q^\mu G_{\mu\nu}^{\mathrm{CDE}} + G_{\mu\nu}^{\mathrm{CDE}} q^\mu\right)\partial^\nu \frac{1}{q^2 - m_1^2} \\
&\quad + \frac{1}{q^2 - m_1^2}\left(q^\mu G_{\mu\nu}^{\mathrm{CDE}} + G_{\mu\nu}^{\mathrm{CDE}} q^\mu\right)\partial^\nu \frac{1}{q^2 - m_1^2}\left(q^\rho G_{\rho\sigma}^{\mathrm{CDE}} + G_{\rho\sigma}^{\mathrm{CDE}} q^\rho\right)\partial^\sigma \frac{1}{q^2 - m_1^2} \\
&\quad - \frac{1}{q^2 - m_1^2}\eta^{\mu\nu} G_{\mu\alpha}^{\mathrm{CDE}} G_{\nu\beta}^{\mathrm{CDE}} \partial^\alpha\partial^\beta \frac{1}{q^2 - m_1^2}.
\end{aligned}
\tag{44}
$$

Now we substitute in the expression of $G_{\mu\nu}^{\mathrm{CDE}}$ in Eq. (27) and truncate the result at operator dimension four:

$$
\begin{aligned}
\Delta_1^{\mathrm{CDE}} &= \frac{1}{q^2-m_1^2} - i F_{\mu\nu}\frac{1}{q^2-m_1^2}q^\mu\partial^\nu\frac{1}{q^2-m_1^2} - \frac{i}{3}\left(P_{\alpha_1}F_{\mu\nu}\right)\frac{1}{q^2-m_1^2}(2q^\mu\partial^{\alpha_1}+\eta^{\mu\alpha_1})\partial^\nu\frac{1}{q^2-m_1^2} \\
&\quad + \frac{1}{4}F_{\mu\nu}F_{\rho\sigma}\left(\eta^{\mu\rho}\frac{1}{q^2-m_1^2}\partial^\nu\partial^\sigma\frac{1}{q^2-m_1^2} - 4\frac{1}{q^2-m_1^2}q^\mu\partial^\nu\frac{1}{q^2-m_1^2}q^\rho\partial^\sigma\frac{1}{q^2-m_1^2}\right) \\
&\quad - \frac{i}{4}\left(P_{\alpha_1}P_{\alpha_2}F_{\mu\nu}\right)\frac{1}{q^2-m_1^2}(q^\mu\partial^{\alpha_2}+\eta^{\mu\alpha_2})\partial^\nu\partial^{\alpha_1}\frac{1}{q^2-m_1^2}.
\end{aligned}
\tag{45}
$$

Next, we expand the factor $U_1^{\mathrm{CDE}}$ using Eq. (26b). Note that $U_1^{\mathrm{CDE}}$ in this example is the last factor in the expression, so all of its momentum derivatives $\partial^{\alpha_i}$ will be acting on the unity function $\mathbb{1}$ to yield zero; only the first term in Eq. (26b) survives, and we can set

$$
U_1^{\mathrm{CDE}} = U_1.
\tag{46}
$$

Substituting Eqs. (45) and (46) into Eq. (43), we obtain

$$
\begin{aligned}
T_1 = -i \int d^d x \int \frac{d^d q}{(2\pi)^d} \, \mathrm{tr}\bigg\{ & U_1\Big[\frac{1}{q^2-m_1^2}\Big] + F_{\mu\nu}U_1\Big[-i\frac{1}{q^2-m_1^2}q^\mu\partial^\nu\frac{1}{q^2-m_1^2}\Big] \\
& + \left(P_{\alpha_1}F_{\mu\nu}\right)U_1\Big[-\frac{i}{3}\frac{1}{q^2-m_1^2}(2q^\mu\partial^{\alpha_1}+\eta^{\mu\alpha_1})\partial^\nu\frac{1}{q^2-m_1^2}\Big] \\
& + F_{\mu\nu}F_{\rho\sigma}U_1\Big[\frac{1}{4}\eta^{\mu\rho}\frac{1}{q^2-m_1^2}\partial^\nu\partial^\sigma\frac{1}{q^2-m_1^2} - \frac{1}{q^2-m_1^2}q^\mu\partial^\nu\frac{1}{q^2-m_1^2}q^\rho\partial^\sigma\frac{1}{q^2-m_1^2}\Big] \\
& + \left(P_{\alpha_1}P_{\alpha_2}F_{\mu\nu}\right)U_1\Big[-\frac{i}{4}\frac{1}{q^2-m_1^2}(q^\mu\partial^{\alpha_2}+\eta^{\mu\alpha_2})\partial^\nu\partial^{\alpha_1}\frac{1}{q^2-m_1^2}\Big]\bigg\}\bigg|_{\mathrm{hard}}.
\end{aligned}
\tag{47}
$$

We see that up to operator dimension six, there are five effective operators:

$$\text{tr}(U_1)\,,\ \text{tr}\big(F_{\mu\nu}U_1\big)\,,\ \text{tr}\big[\big(P_{\alpha_1}F_{\mu\nu}\big)U_1\big]\,,\ \text{tr}\big(F_{\mu\nu}F_{\rho\sigma}U_1\big)\,,\ \text{tr}\big[\big(P_{\alpha_1}P_{\alpha_2}F_{\mu\nu}\big)U_1\big]\,, \qquad (48)$$

each multiplied by a $q$-section factor gathered in the square brackets in Eq. (47).

**Perform loop integrals**

We carry out the $q$-derivatives in the $q$-sections in Eq. (47) and compute the hard region contributions to the loop integrals to obtain the Wilson coefficients. Only two of the five integrals are nonzero; the end result is

$$T_1 = \int \mathrm{d}^d x \ \tfrac{1}{16\pi^2} \ \text{tr}\bigg[m_1^2\Big(1-\log\tfrac{m_1^2}{\mu^2}\Big)U_1 + \tfrac{1}{12m_1^2}\,F_{\mu\nu}F^{\mu\nu}U_1\bigg], \qquad (49)$$

reproducing Eq. (C.1) in Ref. [1].

   This completes the demonstration of the evaluation procedure detailed in Sec. 3.3. Clearly, the same procedure applies to any functional supertrace targeted by STrEAM, with $f\big(P_\mu, \{U_k\}\big)$ in the form of Eq. (14). For more complicated expressions of $f$, the CDE steps shown in Eqs. (44)-(46) are more involved, and the resulting expression in Eq. (47) would contain more effective operators and more complicated $q$-sections. Accordingly, the $q$-derivatives and the loop integral $-i\int\frac{\mathrm{d}^d q}{(2\pi)^d}$ are also more tedious to carry out. Nevertheless, these steps are completely algorithmic. STrEAM automates them and provides results analogous to Eq. (49) as its output. For example, all the other supertraces listed in App. C in Ref. [1] are also readily reproduced with STrEAM. In practical EFT matching calculations, one then substitutes in the explicit expressions for $\{U_k\}$ and evaluates the remaining trace ("tr" in Eq. (49)) as explained in Ref. [1]. One can further translate the result into an EFT operator basis via integration by parts, equations of motion and operator identities if desired.

# 4  STrEAM Manual

In this section, we provide a user manual for STrEAM. We will go over some basics of using this package and show a few concrete examples.

**Downloading and loading**

STrEAM is a Mathematica package publicly available on GitHub ⬡ [34]. Once the file "STrEAM.m" is placed in the user's directory of choice /path/to/package/, it can be loaded in Mathematica with the usual command:

```
In[1]:= <<"/path/to/package/STrEAM.m";
```

**Main functions**

STrEAM is a compact package with essentially a single main function SuperTrace that carries out the procedure summarized in Sec. 3.3. Once the package is loaded, the user can readily execute this main function:

```
In[2]:= SuperTrace[dim, flist]
```

As indicated above, it has two mandatory arguments. dim is an Integer that specifies the desired operator dimension in the evaluation result. flist is a List that specifies the functional operator $f\big(P_\mu, \{U_k\}\big)$ to be traced over; it consists of P$_\mu$, U$_k$, $\Delta_i$, and $\Lambda_i$, organized in the form of Eq. (14). This is the main input to the function, and a few remarks are in order:

- The symbols P, $\Delta$, and $\Lambda$ are reserved in STrEAM, such that the package can recognize these key elements in `flist`. On the other hand, the symbol U is not reserved, and the user can choose their own symbols for the elements $U_k$.

- In `flist`, each $P_\mu$ must come with a subscript as its Lorentz index. One can choose their own symbol for it (but the same Lorentz index should not appear more than twice). The Lorentz indices $\mu_i$ are reserved in STrEAM as dummy indices generated in the expansions as well as in the final outputs. If they were encountered in the input, they would be automatically replaced with $\nu_j$ with some proper integer j.

- Recall from Eq. (12) that each propagator $\Delta_i$ or $\Lambda_i$ must come with a subscript i, indicating its mass $m_i$. These mass labels are non-negative integers, and will be used for specifying the list of heavy masses. This is crucial because STrEAM evaluates the hard region contributions to supertraces. The label "0" is reserved for massless propagators, *i.e.*, we have stipulated $m_0 = 0$ (*cf.* Eq. (12)):

$$\Delta_0 \equiv \frac{1}{P^2}, \qquad \Lambda_0 \equiv \frac{1}{\slashed{P}}. \tag{50}$$

- As explained in Sec. 2, in order to accommodate the log-type supertraces via Eq. (15), we allow the last $U$ factor in $f(P_\mu, \{U_k\})$ to be trivial. This is implemented by allowing the very last $U_k$ factor in `flist` to be absent.

There are also a few options for `SuperTrace`:

| Option | Default | Description |
| --- | --- | --- |
| Udimlist | {1,...,1} | Minimum operator dimensions of $\{U_k\}$ |
| Heavylist | {1} | Heavy mass labels |
| SoftOrd | 0 | Additional power(s) of $m_{\text{light}}/m_{\text{heavy}}$ |
| No$\gamma$inU | False | No Dirac matrices $\gamma_\mu$ in $\{U_k\}$ |
| display | False | Print result |

As explained in Sec. 3.2, the minimum operator dimensions of the $U_k$ factors, gathered in `Udimlist`, are needed to determine when to truncate the CDE. The default setting is that they are all unity. `Heavylist` specifies the list of heavy masses through their (positive integer) labels; this is crucial for identifying the hard region contributions. The default setting is that only $m_1$ is heavy. `SoftOrd` is a non-negative integer. When it is set to the default value 0, the Wilson coefficient of an effective operator with operator dimension $\dim_{\mathcal{O}}$ will be computed up to ($\dim - \dim_{\mathcal{O}}$) powers of $m_{\text{light}}/m_{\text{heavy}}$. If additional powers are desired, one can specify a positive `SoftOrd`. `No$\gamma$inU` can be used to simplify the result when there are no Dirac matrices $\gamma_\mu$ in the $U_k$ factors. The option `display` controls whether to print the result.

Let us consider a very simple example — the supertrace discussed in Sec. 3.4:

$$T_1 \equiv -i \, \text{STr}\left[\Delta_1 U_1^{[2]}\right]\Big|_{\text{hard}}. \tag{51}$$

To evaluate this supertrace up to operator dimension six, one simply runs

In[2]:= `SuperTrace[6, {`$\Delta_1$`, `$U_1$`}, Udimlist`$\to$`{2}]`

SuperTrace returns the evaluation result as a list of terms, with each term in the following form:

$$\{\texttt{coeff, oper, dim}\}$$

where `coeff` and `oper` are lists themselves that contain the Wilson coefficient (multiplied by $16\pi^2$) and the effective operator respectively; `dim` records the operator dimension of the term. For instance, the above example yields an output with two such terms:

Out[2]= $\left\{\left\{\left\{\left\{m_1^2\left(1\text{-}\mathtt{Log}\left[\frac{m_1^2}{\mu^2}\right]\right)\right\}\right\}, \{\{U_1\}\}, 2\right\},\right.$

$\left.\left\{\left\{\left\{\frac{1}{12m_1^2}\right\}\right\}, \{\{F_{\mu_1,\mu_2}\}, \{F_{\mu_1,\mu_2}\}, \{U_1\}\}, 6\right\}\right\}$

When the option `display`$\to$`True` is used, SuperTrace will print the evaluation result in `TableForm`, together with the input supertrace. Again for the example in Eq. (51):

In[3]:= `SuperTrace[6, {`$\Delta_1$`, `$U_1$`}, Udimlist`$\to$`{2}, display`$\to$`True];`

will print

$$-\mathtt{iSTr}\left[\frac{1}{\mathtt{P}^2\text{-}m_1^2}U_1\right]|_{\mathtt{hard}} \;=\; \int d^4x \;\; \frac{1}{16\pi^2} \;\; \mathtt{tr}\{$$

| | | |
|---|---|---|
| $m_1^2\left(1\text{-}\mathtt{Log}\left[\frac{m_1^2}{\mu^2}\right]\right)$ | $(U_1)$ | (dim-2) |
| $\frac{1}{12m_1^2}$ | $(F_{\mu_1\mu_2})(F_{\mu_1\mu_2})(U_1)$ | (dim-6) |

$$\}$$

We recommend using this option for checking the result in a more readable format.

There is also an alternative function SuperTraceFromExpr, which is a slight variant of SuperTrace that takes an expression `fexpr`, instead of a `List` `flist`, as the input for specifying the functional operator $f\left(P_\mu, \{U_k\}\right)$:

In[4]:= `SuperTraceFromExpr[dim, fexpr]`

It has the same set of options as SuperTrace. The input expression `fexpr` is obtained by putting together the elements in `flist` with `NonCommutativeMultiply` (`**`). For instance, to evaluate the same example in Eq. (51), one can execute

In[4]:= `SuperTraceFromExpr[6, `$\Delta_1$`**`$U_1$`, Udimlist`$\to$`{2}, display`$\to$`True];`

The output of SuperTraceFromExpr is the same as that of SuperTrace.

### Reserved variables

As mentioned before, in STrEAM the symbols P, $\Delta$, and $\Lambda$ are reserved for input recognition. The Lorentz indices $\mu_i$ are reserved for calculation purposes. P and $\mu_i$ are also used in the outputs. Apart from these, the following symbols are also reserved for special meanings:

| Reserved symbol | Meaning |
|---|---|
| m | Particle masses $m_i$ |
| d | Spacetime dimension in dimensional regularization |
| $\eta$ | Spacetime metric $\eta_{\mu\nu} = \text{diag}\,(1,-1,-1,-1)$ |
| $\varepsilon$ | Levi-Civita symbol $\varepsilon_{\mu\nu\rho\sigma}$ with $\varepsilon_{0123} = -1$ |
| F | Field strength $F_{\mu\nu} \equiv -i\left[P_\mu, P_\nu\right] = g_a\, G^a_{\mu\nu}\, T^a$ |
| $\gamma$ | Dirac matrices $\gamma_\mu$ |
| $\sigma$F | $\sigma^{\mu\nu} F_{\mu\nu}$ in the dipole factor (see Eq. (34)) |
| Pslash | $\not{P} = \gamma^\mu P_\mu$ |
| LoopI | Loop integral in Eq. (38) for $\geq 3$ heavy masses |

**Additional examples**

Finally, we show a few selected input examples to better illustrate the syntax. We will not include their outputs or prints here. These results, as well as more demonstration examples, are collected in a Mathematica notebook "STrEAM_examples.nb" ⟳ [34]. For each example below, one can add the option display→True if desired.

- Supertraces converted from log-type via Eq. (15),

$$-i\,\text{STr}\left[\frac{1}{P^2 - m_1^2}\right]\Bigg|_{\text{hard}}, \qquad -i\,\text{STr}\left[\frac{1}{\not{P} - m_1}\right]\Bigg|_{\text{hard}}, \tag{52}$$

can be evaluated up to operator dimension six with

```
In[5]:= SuperTrace[6, {Δ₁}]

In[6]:= SuperTrace[6, {Λ₁}, NoγinU→True]
```

Note that we have turned on the option NoγinU→True for the fermionic one.

- A supertrace with both heavy and light propagators ($m_1$ and $m_2$), such as

$$-i\,\text{STr}\left[\frac{1}{P^2 - m_1^2}\, U_1^{[1]}\, \frac{1}{P^2 - m_2^2}\, U_2^{[1]}\, \frac{1}{P^2 - m_1^2}\, U_3^{[1]}\, \frac{1}{P^2 - m_2^2}\, U_4^{[1]}\right]\Bigg|_{\text{hard}}, \tag{53}$$

can be evaluated up to operator dimension six with

```
In[7]:= SuperTrace[6, {Δ₁, U₁, Δ₂, U₂, Δ₁, U₃, Δ₂, U₄}]
```

or equivalently with

```
In[7]:= SuperTraceFromExpr[6, Δ₁**U₁**Δ₂**U₂**Δ₁**U₃**Δ₂**U₄]
```

- A supertrace with explicit open covariant derivatives, such as

$$-i\,\mathrm{STr}\left[\frac{1}{P^2-m_1^2}\,U_1^{[1]}\frac{1}{P^2-m_2^2}\,P_\mu Z^{\mu[1]}\frac{1}{P^2}\,U_3^{[2]}\frac{1}{P^2-m_2^2}\,U_4^{[1]}\right]\bigg|_{\mathrm{hard}},\qquad(54)$$

can be evaluated up to operator dimension six with

```
In[8]:= SuperTrace[6, {Δ₁, U₁, Δ₂, Pᵥ, Zᵥ, Δ₀, U₃, Δ₂, U₄},
            Udimlist→{1, 1, 2, 1}]
```

- A supertrace with fermionic propagators, such as

$$-i\,\mathrm{STr}\left[\frac{1}{P^2-m_1^2}\,U_1^{[1]}\frac{1}{P^2-m_2^2}\,U_2^{[3/2]}\frac{1}{\not{P}}\,U_3^{[3/2]}\frac{1}{P^2-m_2^2}\,U_4^{[1]}\right]\bigg|_{\mathrm{hard}},\qquad(55)$$

can be evaluated up to operator dimension six with

```
In[9]:= SuperTrace[6, {Δ₁, U₁, Δ₂, U₂, Λ₀, U₃, Δ₂, U₄},
            Udimlist→{1, 3/2, 3/2, 1}]
```

## Acknowledgments

We thank Javier Fuentes-Martín, Matthias König, Julie Pagès, Anders Eller Thomsen, and Felix Wilsch for communications about their related work [50] and cross-checks. T.C. and X.L. are supported by the U.S. Department of Energy, under grant number DE-SC0011640. Z.Z.'s work was supported in part by the U.S. Department of Energy, Office of Science, Office of High Energy Physics, under Award Number DE-AC02-05CH11231.

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
