# Peer review of "STrEAMlining EFT Matching"

_SciPost Physics, doi:SciPost Phys. 10, 098 (2021)_

## Round 1 · Referee Report · Anonymous (Referee 1) · 2021-1-22

Report

This paper introduces STrEAM, a Mathematica package for the calculation of functional supertraces. It provides a clear presentation of the internal algorithm and instructions for the user. STrEAM performs a step in the functional matching of a UV theory to an effective field theory, a common calculation in high-energy physics. This renders the package a useful tool in this area. Because of this, I would recommend publication, after the requested changes are addressed.

Requested changes

  1. It is not clear from the paper whether or not STrEAM can be used to do the matching for UV theories containing interactions with derivatives of the heavy fields. If the absence of such derivatives is an assumption, it should be stated clearly. Otherwise, it would be an impressive feature that should be stressed.

  2. In footnote 1, it is mentioned that another program with similar purpose, SuperTracer, is available. The potential user would benefit from a comment on the differences and similarities between the two.

  3. Towards the end of section 4, some examples are given of how to use STrEAM. Since the main use case of STrEAM is for matching calculations, it would also help the user to briefly present a UV model for which the matching to the effective theory involves computing these supertraces, and to comment on how to proceed after using STrEAM to complete the calculation. While it is true that Ref. [1]'s purpose is to provide these details, having a short explanation in place would be appropriate. At the minimum, a pointer to the revelent part of in Ref. [1] would be useful.

---

## Round 1 · Referee Report · Anonymous (Referee 2) · 2021-2-8

Report

In this paper, the authors provide and present a Mathematica package (STrEAM), which automates the streamlined functional matching prescription that they previously showed in arXiv:2011.02484.
The goal of the procedure is the integration out, up to one-loop level, of any UV perturbative state, regardless the structure of its interactions, including derivative couplings. Applying functional method, the matching of an UV theory into an EFT reduces to the enumeration and evaluation of functional supertraces, as summarized and emphasized in this paper. STrEAM implements the covariant derivative expansion (CDE), which is synthetically but well reviewed in the paper, in order to calculate supertraces. In this approach, furthermore, no prior determination of the EFT operator basis is required: effective operators as well as the corresponding Wilson coefficients are obtained as outputs. In the presented streamlined prescription it is also possible to perform the matching up to any operator dimension in the EFT expansion. Among the automated tools for EFT calculations, STrEAM is the first public package for general one-loop matching, going beyond dimension six level as well.

Therefore, the procedure provided and presented by the authors allows large generality, with few steps (one of which is accomplished by STrEAM) and a minimal and clear algorithm. The paper is well written and organized and contains a manual for the STrEAM package. The prescriptions and the tool presented in this paper are therefore of interest for community and I would recommend the publication.

I have, however, a couple of suggestions for the authors.

In my view, one of the greatest strengths of the streamlined prescription shown here lies in the possibility of going beyond dimension six operators, which is a new feature in the automated tool STrEAM. However, in all the examples shown at the end of section 4, supertraces are evaluated up to dimension six. This may partly hide the STrEAM ability to perform the matching at any order in the EFT expansion, on which I would put more emphasis.

The authors mention in a footnote that the program SuperTraces has been released simultaneously to STrEAM. It might be useful for the reader if they could briefly explain if there are significant differences among the two tools.

---

## Round 1 · Referee Report · Anonymous (Referee 3) · 2021-3-1

Strengths

1) Clear presentation

2) Useful code for practical functional matching calculations

Weaknesses

1) Examples could be expanded, in particular including fermions or open covariant derivatives

2) The discussion on open covariant derivatives in Section 3.1 could potentially be confusing to those unfamiliar with the CDE, since it talks about only closed covariant derivatives being non-zero in the "simplified CDE" and removes open covariant derivatives in the "original CDE", which seems to contradict the earlier point in Section 2 that open covariant derivatives are part of the scope and one of the advantages of this code. Of course the latter Section is referring to the terms in the expansion of the general functional operator while the former is the form of the general functional operator itself, but it may be worth mentioning this explicitly in the CDE review.

Report

This paper presents a mathematica code for automating the evaluation of traces in functional one-loop matching, following recent simplifications in performing such computations by the authors and others. The novel results are in a separate paper (Ref. 1) and this work reads like an extended appendix, but the code itself is a sufficiently useful result to recommend publication here.

---

## Editorial Decision

published